# Signaling Cross-Talk between Salicylic and Gentisic Acid in the ‘*Candidatus* Phytoplasma Solani’ Interaction with Sangiovese Vines

**DOI:** 10.3390/plants12142695

**Published:** 2023-07-19

**Authors:** Eliana Nutricati, Mariarosaria De Pascali, Carmine Negro, Piero Attilio Bianco, Fabio Quaglino, Alessandro Passera, Roberto Pierro, Carmine Marcone, Alessandra Panattoni, Erika Sabella, Luigi De Bellis, Andrea Luvisi

**Affiliations:** 1Department of Biological and Environmental Sciences and Technologies, University of Salento, Via Provinciale Lecce-Monteroni, 73100 Lecce, Italy; eliana.nutricati@unisalento.it (E.N.); mariarosaria.depascali@unisalento.it (M.D.P.); erika.sabella@unisalento.it (E.S.); luigi.debellis@unisalento.it (L.D.B.); andrea.luvisi@unisalento.it (A.L.); 2Department of Agricultural and Environmental Sciences, Production, Landscape, Agroenergy (DiSAA), University of Milan, Via Celoria 2, 20133 Milano, Italy; piero.bianco@unimi.it (P.A.B.); fabio.quaglino@unimi.it (F.Q.); alessandro.passera@unimi.it (A.P.); 3Department of Pharmacy, University of Salerno, Via Giovanni Paolo II, 132, 84084 Fisciano, Italy; rob.pierro@outlook.it (R.P.); cmarcone@unisa.it (C.M.); 4Department of Agriculture, Food and Environment (DAFE), University of Pisa, Via del Borghetto 80, 56124 Pisa, Italy; alessandra.panattoni@unipi.it

**Keywords:** *Vitis vinifera* L., grapevine diseases, phenylpropanoids, pathogenesis-related proteins

## Abstract

“Bois noir” disease associated with ‘*Candidatus* Phytoplasma solani’ seriously compromises the production and survival of grapevines (*Vitis vinifera* L.) in Europe. Understanding the plant response to phytoplasmas should help to improve disease control strategies. Using a combined metabolomic and transcriptomic analysis, this work, therefore, investigated the phytoplasma–grapevine interaction in red cultivar Sangiovese in a vineyard over four seasonal growth stages (from late spring to late summer), comparing leaves from healthy and infected grapevines (symptomatic and symptomless). We found an accumulation of both conjugate and free salicylic acids (SAs) in the leaves of ‘*Ca*. P. solani’-positive plants from early stages of infection, when plants are still asymptomatic. A strong accumulation of gentisic acid (GA) associated with symptoms progression was found for the first time. A detailed analysis of phenylpropanoids revealed a significant accumulation of hydroxycinnamic acids, flavonols, flavan 3-ols, and anthocyanin cyanidin 3-*O*-glucoside, which are extensively studied due to their involvement in the plant response to various pathogens. Metabolomic data corroborated by gene expression analysis indicated that phenylpropanoid biosynthetic and salicylic acid-responsive genes were upregulated in ‘*Ca*. P. solani-positive plants compared to -negative ones during the observed period.

## 1. Introduction

Phytoplasmas are plant pathogenic bacteria in the class Mollicutes transmitted by phloem sap-feeding insects. They have a broad range of hosts among plant species worldwide, including many economically important crops such as grapevine, fruit trees, and ornamental plants [1,2]. In Europe, ‘*Candidatus* Phytoplasma solani’ (‘*Ca*. P. solani’), taxonomic subgroup 16SrXII-A [3] is associated with “Bois noir” (BN) disease, which can lead to serious losses of grape clusters [4]. BN is endemic in the Euro-Mediterranean area, and it is characterized by a disease cycle, including insect vectors and many herbaceous plants as phytoplasma reservoirs [5,6]. Phytoplasma are still poorly characterized plant pathogens because of their low concentration in plants and the difficulties of in vitro cultivation. Agricultural interest has arisen from the possibilities of controlling phytoplasma diseases through new effective strategies.

Although phytoplasmas interfere with plant physiological processes [7,8], the biochemical and molecular mechanisms responsible for the interaction with their host remain uncertain. However, some studies on the phytoplasma genome have identified the presence of phytoplasma effectors, such as SAP11, SAP54, and TENGU, describing their role in molecular interactions with plant hosts. These effectors modify the activity of plant transcription factors, regulating the expression of genes involved in several metabolic pathways [9,10,11].

Traditional biochemical studies, transcriptomic, proteomic, and metabolomic approaches have been used to study plant–phytoplasma interactions, highlighting the alterations of many metabolic pathways [12,13], above all, sugars [14] and flavonoid biosynthesis [15] in phytoplasma-infected plants.

Plants have complex defense systems to counteract pathogens. When they perceive pathogen signals, they immediately react and induce a complex signaling mechanism, which enhances the expression of plant defense-related genes and the synthesis of phenylpropanoids [16].

There is growing evidence that the elicitation of a specific and appropriate plant response to a pathogen requires the integration and coordination of multiple signaling pathways regulated by plant growth regulators, mainly salicylic acid (SA), jasmonic acid (JA), and ethylene (ET) [17].

SA-mediated defense appears to be a common strategy against phytoplasmas [18]: a high level of SA was detected in apple trees infected by ‘*Candidatus* Phytoplasma mali’ [19]. Many SA-related genes were found to be highly expressed upon phytoplasma infection in tomato, grape, and periwinkle [12,20,21]. For example, infected periwinkle plants showed a high level of SA and the upregulation of *PR1* (*pathogenesis-related protein 1*), a marker gene of SA signaling [18]. SA also enhances the expression of genes involved in flavonoid metabolism and the accumulation of non-enzymatic antioxidant compounds, such as hydroxycinnamic acids and flavonols [22]. 

Studies on grapevine responses to phytoplasmas, carried out during the late growing season when grapevine shows disease symptoms, have focused on the transcriptomic as well as metabolomic analyses of infected plants compared with healthy ones [13,15,23,24,25,26]. Overall, the data have revealed the modulation of genes coding for proteins involved in various metabolic pathways, and the upregulation of defense-related genes, such as *NPR1* (*non-expressor of pathogenesis-related protein 1*), coding for an SA receptor, and *PR1*, *PR2,* and *PR5* (*pathogenesis-related proteins 1, 2, and 5*), commonly used as molecular markers for SA signaling in systemic-acquired resistance (SAR) [27,28]. Plants produce high local concentrations of SA during the hypersensitive response (HR), leading to host tissue collapse to limit resources for pathogens, and develop systemic-acquired resistance, which is a form of long-term immune memory [29]. In plants, the hormone has been found in both free and conjugated forms and different derivatives can be produced through hydroxylation, amino acid conjugation, and methylation. These modifications inactivate SA, fine-tuning its activity and making up a temporary, readily available SA storage [30].

Several SA-derived compounds have been identified; however, the mode of action of SA-related metabolites is not completely understood [31]. The main SA derivatives associated with the response to several pathogens are 2,5-dihydroxybenzoic acid (known as gentisic acid) and 2,3-dihydroxybenzoic acid [32].

Gentisic acid (GA) is present in considerable amounts in gentiana (*Gentiana* spp.), grapes (*Vitis vinifera*), citrus fruit (*Citrus* spp.), olives (*Olea europaea*), and sesame (*Sesamum indicum*). Many studies have reported positive effects of GA on human health, such as its antioxidant, neuroprotective, and antimicrobial activities, and therefore, it has been proposed for the treatment of many diseases [33].

The few data available on the accumulation of SA and its glycosylated forms or derivatives in grapevine leaves infected by ‘*Ca.* P. solani’ [13,26] or Flavescence dorée phytoplasma [34] have revealed a higher content in infected leaves compared with healthy ones. However, no studies have been reported to date on the accumulation of GA in grapevines infected by phytoplasmas, except for Prezelj et al. [34] and Rotter et al. [35]. The latter authors reported data regarding the overexpression of the *DMR6* gene showing 69% of identity to the sequence of *Arabidopsis AtDMR6* gene, coding for a SA 5-hydroxylase (E.C. 1.14.13.1) [36].

In addition, an SA 3-hydroxylase, which catalyzes the formation of 2,3-DHBA, seems to play a pivotal role in SA catabolism and homeostasis during the expression of pathogen resistance [31] and leaf senescence [37].

In the present work, we extend our previous study [38] by exploring over four seasonal growth stages the accumulation and physiological role of SA and its derivatives, GA and 2,3-DHBA, in a compatible interaction between ‘*Ca*. P. solani’ and Sangiovese, a susceptible red grape from central Italy. The metabolomic analysis included the evaluation of changes in phenylpropanoid, quantifying the phenolics, flavonols, anthocyanins, and proanthocyanidins through high-performance liquid chromatography–mass spectrometry (HPLC-MS). A transcriptomic analysis showed the expression of genes involved in the phenylpropanoid biosynthesis and, among several *PR* genes, *PR1*, *PR2,* and *PR5*.

Our findings highlight the importance of SA- and GA-mediated defense against ‘*Ca*. P. solani’ in grapevine and offer insights into grapevine–phytoplasma interactions to develop new strategies for controlling BN disease.

## 2. Results

### 2.1. Plant Symptoms

Leaves from ‘*Ca*. P. solani’-positive and -negative plants were sampled in four periods, from the late spring to late summer (May, July, August, and September) of 2020. In May and July, samples were collected from ‘*Ca*. P. solani’-positive plants showing symptoms of severity class 0 (no symptoms observed). On the other hand, in August, ‘*Ca*. P. solani’-positive plants were classified as belonging to severity class 2 (the symptoms were mild), and lastly, in September, the plants selected for analysis were classified as belonging to severity class 3 (more than three shoots with reddening leaves) [39]. Therefore, samples collected in May and July from ‘*Ca*. P. solani’-positive plants were considered as asymptomatic, while those collected in August and September were considered as symptomatic. Leaves collected from ‘*Ca*. P. solani’-negative plants were classified as class 0 (no symptoms observed) during all sampling stages.

### 2.2. Characterization of Phenylpropanoids Accumulated in Phytoplasma Infected Plants

#### 2.2.1. Qualitative Analysis

Table 1 reports the phenolic compounds identified by negative ionization mode using HPLC/MS/TOF in leaves of ‘*Ca*. P. solani’-positive or -negative plants. All compounds were detected, although in different amounts, in both positive and negative plants.

Most of the compounds were present in either glucoside or glucuronide forms, as previously reported in literature [40,41,42]. We found different benzoic acid derivatives, such as dihydroxybenzoic acid glucoside isomer 1(1A); gentisic acid (2A) and dihydroxybenzoic acid glucoside isomer 2 (3A); 2,3 dihydroxybenzoic acid (5A); hydroxybenzoic glucoside (6A); and salicylic acid (15A).

Among the phenylpropanoids, caffeic acid glucoside (4A), coumaric acid glucoside (8A), and ferulic acid glucoside (10A) were identified.

Among the flavonoids, we detected flavan-3-ols, such as catechin glucoside (11A) and epicatechin glucoside (13A); different conjugated forms of flavanol myricetin, such as myricetin 3-*O*-glucuronide (12A) and myricetin 3-*O*-glucoside (14A); quercetin 3-*O*-glucoside (16A), quercetin 3-*O*-glucuronide (17A), and quercetin 3-*O*-rhamnoside (21A); and kaempferol 3-*O*-glucoside (18A), kaempferol 3-*O*-rutinoside (19A), and kaempferol 3-*O*-glucuronide (20A). Extracts also contained resveratrol glucoside (22A).

To verify the presence of cyanidin 3-*O*-glucoside, which was detected in our previous study [38] in leaves of ‘*Ca*. P. solani’-positive plants, all extracts were also analyzed by positive ionization mode, confirming the presence of cyanidin 3-*O*-glucoside (24A) and two other anthocyanins, i.e., delphinidin 3-*O*-glucoside (23A) and peonidin 3-*O*-glucoside (25A).

**Table 1 plants-12-02695-t001:** Putative identification of main phenolic compounds extracted from leaves collected from *Vitis vinifera* cv. Sangiovese ‘*Ca*. P. solani’-positive and -negative plants detected by high-performance liquid chromatography time-of-flight mass spectrometry HPLC ESI/MS-TOF.

**No.**	**Compound**	**RT ^a^ (min)**	***m/z* exp. ^b^**	***m/z* calc. ^c^**	**(M-H)^−^**	**Error (ppm)**	**Reference**
**1A**	Dihydroxybenzoic acid-glucoside is. 1	2.626	315.0736	315.0722	C_13_H_15_O_9_	−4.44	[43,44,45]
**2A**	Gentisic acid *	3.703	153.0194	153.0193	C_7_H_5_O_4_	−0.65	
**3A**	Dihydroxybenzoic acid-glucoside is. 2	4.500	315.0754	315.0733	C_13_H_15_O_9_	−10.15	[44,46]
**4A**	Caffeic acid glucoside	5.129	341.0896	341.0878	C_15_H_17_O_9_	−5.73	[47]
**5A**	2,3 Dihydroxybenzoic acid *	5.184	153.0208	153.0193	C_7_H_5_O_4_	−9.80	
**6A**	Hydroxybenzoic acid glucoside	5.827	299.0789	299.0772	C_13_H_16_O_8_	−5.68	[44]
**7A**	Catechin *	5.889	289.0721	289.0718	C_15_H_13_O_6_	−1.03	
**8A**	Coumaric-3-*O*- acid glucoside	6.374	325.0940	325.0920	C_15_H_17_O_8_	−6.15	[43,45]
**9A**	Epicatechin *	6.952	289.0737	289.0718	C_15_H_13_O_6_	−6.57	
**10A**	Ferulic acid-3-*O*-glucoside	7.204	355.1049	355.1035	C_16_H_20_O_9_	−3.94	[48]
**11A**	Catechin-3-*O*-glucoside	7.253	451.1324	451.1318	C_21_H_23_O_11_	−1.33	[43]
**12A**	Myricetin-3-*O*-glucuronide	8.796	493.0640	493.0624	C_21_H_17_O_14_	−3.24	[46]
**13A**	Epicatechin- 3-*O*-glucoside	8.819	451.1278	451.1318	C_21_H_23_O_11_	8.86	[43]
**14A**	Myricetin-3-*O*-glucoside	8.845	479.0842	479.0831	C_21_H_19_O_13_	−2.29	[45]
**15A**	Salicylic acid *	9.426	137.0248	137.0244	C_7_H_5_O_3_	−3.04	
**16A**	Quercetin-3-*O*-glucoside *	9.488	463.0908	463.0882	C_21_H_19_O_12_	−5.61	[46]
**17A**	Quercetin 3-*O*-glucuronide	9.940	477.0726	477.0675	C_21_H_17_O_13_	−10.69	[46]
**18A**	Kaempferol 3-*O*-glucoside *	10.438	447.0960	447.0933	C_21_H_19_O_11_	−6.03	[46]
**19A**	Kaempferol 3-*O*-rutinoside	10.637	593.1513	593.1512	C_27_H_29_O_15_	−0.16	[47]
**20A**	Kaempferol 3-*O*-glucuronide	10.808	461.0750	461.0725	C_21_H_17_O_12_	−5.42	[46]
**21A**	Quercetin 3-*O*-rhamnoside	10.819	447.0968	447.0933	C_21_H_19_O_11_	−7.82	[47]
**22A**	Resveratrol 3-*O*-glucoside *	11.167	389.1253	389.1242	C_20_H_21_O_8_	−2.82	[47]
**No.**	**Compound**	**RT ^a^ (min)**	***m/z* exp. ^b^**	***m/z* calc. ^c^**	**(M-H)^+^**	**Error (ppm)**	**Reference**
**23A**	Delphinidin 3-*O*-glucoside *	8.357	465.1033	465.1028	C_21_H_21_O_12_	−1.07	[49]
**24A**	Cyanidin 3-*O*-glucoside *	9.637	449.1114	449.1078	C_21_H_21_O_11_	−8.01	[49]
**25A**	Peonidin 3-*O*-glucoside *	12.241	463.1253	463.1235	C_22_H_23_O_11_	−3.88	[49]

^a^ Retention time (min), ^b^ *m*/*z* experimental, ^c^ *m*/*z* calculated from software Agilent Mass Hunter 7.0. * Compound verified by a comparison with authentical chemical standard.

#### 2.2.2. Identification of Phenylpropanoid Compounds

Because many of the phenylpropanoids are detected in glycosylated form, to confirm and quantify each compound identified, extracts from leaves collected from ‘*Ca*. P. solani’ -positive and -negative plants were digested by β-glucosidase. The typical chromatogram is presented in Figure 1.

Table 2 shows the compounds identified after the enzymatic treatment: mainly two dihydroxybenzoic acids, the 2,5 dihydroxybenzoic acid (or GA), the 2,3 dihydroxybenzoic acid, (compounds **1B** and **2B,** respectively), and salicylic acid (compound **8B**, o-hydroxybenzoic acid).

We also confirmed the presence of hydroxycinnamic acids, i.e., p-coumaric, caffeic and ferulic acids, as well as the aglycon form of flavonoids quercetin, myricetin, kaempferol, and the flavan-3-ol catechin and epicatechin.

### 2.3. Phenolic Compounds Accumulate in ‘Ca. P. solani’-Positive Plants

To evaluate the changes in phenylpropanoid contents with regard to BN health status, a quantitative analysis was carried out using chemical standards.

In the chromatogram (Figure 1), the peak 1B corresponds to gentisic acid, resulting from its coelution with standard 2,5 DHBA under the same chromatographic conditions and by comparing its retention time with those previously reported [50]. Figure 2 shows the mass spectra of the compound obtained by ESI in negative ion mode and their UV/Vis peaks. The mass spectrum from the total ion current chromatogram showed a main fragment, which corresponded with the molecular weight of gentisic acid. The mass spectrum also showed an ion at *m*/*z* 108, probably arising from the decarboxylation of gentisic acid. The relative intensities of the mass peaks from plant extracts precisely matched those of the standard 2,5 DHBA.

A comparison of the characteristics of compound no. **8B** with the MS-spectrum of SA (as standard) showed a perfect match, thus confirming the identity of the two compounds (Figure 3).

Similarly, peak no. 2B was coeluted with standard 2,3 DHBA, and showed a retention time identical to the peak of the compound found by Zhang et al. [37] (Appendix A).

To investigate the accumulation of phenolic acids in free and conjugated forms, we carried out a time-course quantitative analysis, comparing leaf extracts collected from ‘*Ca*. P. solani’-positive or -negative plants and with different symptom levels according to the different seasons (Figure 4). 

We followed the analysis pattern of free GA and SA along the disease progression (Figure 4), which revealed that the presence of free GA was very low in spring and early summer in leaves from ‘*Ca*. P. solani’-negative plants, similarly to that in the positive ones. However, free GA underwent a slight increase in positive plants in August (0.27 μg/g FW), while in September, it reached a level of about 44 times higher than in negative plants (Figure 4A).

The free SA basal content, in leaves collected from negative plants, was constant throughout the entire growth season, with an average value of about 0.23 μg/g FW. In positive plants, the SA level increased from spring to early summer compared to healthy ones, and in September, free SA increased roughly threefold, reaching the GA content in the same sample (Figure 4B).

In ‘*Ca*. P. solani’-positive plants, a progressive accumulation of conjugated GA was observed, while in ‘*Ca*. P. solani’-negative plants, its level remained nearly constant from late spring to early summer, but tripled in August, and finally returned to the initial level in September (Figure 4A). Conversely, in ‘*Ca*. P. solani’-positive plants, the conjugated GA content gradually increased from May to August, in September, when disease symptoms were pronounced, reaching a high value (about 200 μg/g FW), which was about 25-fold higher than -negative plants and more than 100-fold higher than the GA free amount in the same sample (Figure 4A,C).

The data on conjugated SA (Figure 4D) showed a constant low level in ‘*Ca*. P. solani’-negative plants (class = 0) for the entire analysis period, and a slightly higher level (average value about 4 μg/g FW) in leaves of ‘*Ca*. P. solani’-positive plants from May (asymptomatic plants, class = 0) to August. This was the period when the leaves showed mild BN disease symptoms (class = 2), reaching a maximum level of 14 μg/g FW (about three times higher than the basal level) in September, when the plants showed evident disease symptoms (class = 3). Interestingly, in September, in the leaves of positive plants, the level of conjugated GA was more than nine times higher than conjugated SA.

The other SA catabolite, 2,3 DHBA, was not found in free form and, after β-glucosidase digestion, it was undetected in the leaves of both ‘*Ca*. P. solani’-positive and -negative plants in the late spring and in early summer. On the other hand, it was found in ‘*Ca*. P. solani’-positive samples at a low level in August (about 2.9 μg/g FW) and increased, about threefold, in infected plants in September (Figure 5A). In ‘*Ca*. P. solani’-negative plants, levels of 2,3 DHBA in mid and late summer were very low compared to positive plants.

It is known that hydroxycinnamic acids and monocyclic phenylpropanoids are involved in the interactions between pathogens and host plants. When methanol extracts from ‘*Ca*. P. solani’-positive and -negative grapevine plants were digested with β-glucosidase, p-coumaric and ferulic acid were clearly detected. Both p-coumaric and ferulic acid showed a similar trend during seasonal growth (Figure 5B,C); the increase in hydroxycinnamic acids occurred in parallel with the development and severity of symptoms. The p-coumaric acid levels (Figure 5B) measured in late spring and early summer were either low or absent in both negative and positive plants, whereas in August, the amount increased for both positive and negative plants but without a significant difference. In the last growth stage, p-coumaric levels showed a consistent increase (up to about 100 μg/g FW) in ‘*Ca*. P. solani’-positive plants, whilst in ‘*Ca*. P. solani’-negative plants, the levels decreased approximately to 5 μg/g FW. Ferulic acid started to accumulate in August in both ‘*Ca*. P. solani’-positive and -negative plants, and it reached a peak (33 μg/g FW) only in positive plants in September. In ‘*Ca*. P. solani’-negative plants, ferulic acid was undetectable (Figure 5C).

Flavan-3-ols, catechin and epicatechin, identified in all analyzed samples, maintained throughout the seasons approximately the same concentration in the leaves of negative plants. In May and July, no significant differences between ‘*Ca*. P. solani’-negative and -positive plants were detected, whilst infected plants showed, in August, a 2-fold and 1.5-fold higher content of catechin and epicatechin, respectively, than negative plants. A significant increase (about 3.8-folds) was found in extracts of leaves from ‘*Ca*. P. solani’-positive plants for catechin and about threefold for epicatechin in September (Figure 6A,B).

Flavonols are typical compounds of grapevine leaves. We found three main forms of conjugated quercetin: the most represented molecule was quercetin 3-*O*-glucuronide (Figure 6C). Its trend was similar throughout the growth season in both ‘*Ca*. P. solani’-negative and -positive plants. The quercetin glucoside level (Figure 6D) was similar in negative plants compared to positive ones with a maximum value recorded in August and September for both health statuses. The quercetin 3-*O*-rhamnoside content (Figure 6E) was the lowest and was the only compound with higher levels in spring in negative plants compared to positive plants. In May, it was twofold higher in ‘*Ca*. P. solani’-negative plants compared to -positive ones, whereas in early and mid-summer, the amount decreased and then increased again in late summer. The trend in ‘*Ca*. P. solani’-positive plants was more linear, with low levels in spring and early summer and a significant increase in mid and late summer (Figure 6D). In addition, quercetin 3-*O*-rhamnoside reached similarly high levels for both health statuses in late summer, increasing earlier in ‘*Ca*. P. solani’-positive plants, as it reached a high level (3-folds higher than negative plants) from mid-summer.

Another key flavonol, kaempferol, was predominantly present as kaempferol 3-*O*- glucoside (Figure 6F,G), showing an increase only in ‘*Ca*. P. solani’-positive plants in late summer. Kaempferol 3-*O*-glucuronide instead maintained a relatively constant amount during all sampling periods, regardless of the health status.

Among anthocyanins, we identified cyanidin 3-*O*-glucoside in the leaf extracts. As shown in Figure 6H, there was no significant change in its content with regard to health status from spring to mid-summer, whereas in line with complete leaf reddening, its concentration reached a peak value (about 58 μg/g FW) in ‘*Ca*. P. solani’-positive plants.

### 2.4. Expression Analysis

#### 2.4.1. Modulation of Phenylpropanoid Biosynthetic Pathway Genes

We performed a qPCR assay on a subset of genes involved in phenylpropanoids biosynthesis, including *phenylalanine ammonia-lyase* (E.C. 4.3.1.5., *PAL*), coding the first key enzyme in the phenylpropanoid pathway as: “early genes” (*cinnamate 4-hydroxylase*, E.C. 1.14.14.91, *C4H*; *chalcone synthase* E.C.2.3.1.74, *CHS*; *flavanone-3-hydroxylase* E.C. 1.14.11.9, *F3H*; *flavonoid-3′-hydroxylase* E.C. 1.14.14.82, *F3′H*; and *flavonol synthase*, 1.14.20.6, *FLS*; *dihydroflavonol reductase*, 1.1.1.219, *DFR*) (Figure 7) and “late genes” (*leucoanthocyanidin dioxygenase*, E.C. 1.14.20.4, *LDOX*; *UDP-glucose flavonoid* 3*-O-*glucosyltransferase, E.C. 2.4.2.115, *UFGT*; and *leucoanthocyanidin reductase*, EC 1.17.1.13, *LAR*) of the flavonoid biosynthetic pathway (Figure 8). We assumed that genes were significantly upregulated or downregulated only in the case of a fold change (FC) greater than or equal to 2 or less than equal to −2, respectively.

At the transcriptional level, compared with healthy plants, phytoplasma infection significantly upregulated the expression of genes coding for PAL and C4H in the mid and late summer (August and September) (Figure 7).

For *CHS1* and *CHS2*, we registered an overexpression of about a 14- and 38-fold change in positive plants compared to negative ones in August (Figure 7), respectively, while *CHS3* transcript was upregulated (about 4-fold change) in September.

The two isoforms of *F3H* (*F3H1* and *F3H2*) coding flavanone-3-hydroxylase involved in dihydrokaempferol synthesis showed a similar behavior. In May and July, *F3H1* and *F3H2* transcripts showed no significant changes, but in August, both genes reached the highest expression (about a 12- and 60-fold change, respectively); in September, the *F3H2* transcript overexpression in positive plants was approximatively a 5-fold change. 

*F’3H* coding a flavonoid-3′-hydroxylase showed no significant change in expression with regard to health status.

A high expression level of the *FLS* gene, involved in the biosynthesis of three main flavonols, quercetin, kaempferol, and myricetin, were observed (Figure 8) in Ca. P. solani- positive plants from mid-summer with a 9-fold change in August and a 19-fold change in September.

*DFR* coding a NADPH-dependent dihydroflavonol reductase for leucoanthocyanidins synthesis showed a high expression, 22- and 43-fold (Figure 7), respectively, in August and September in positive plants.

Among the “late genes”, specific for the anthocyanin’s biosynthesis, *LDOX,* which is specific for the conversion of leucoanthocyanidins to anthocyanidins, was overexpressed in September in positive plants (about a 3-fold change) (Figure 8). We also registered an increase in the *UF3GT* transcript, coding a 3-*O*-glucosyltransferase, which is involved in the synthesis of cyanidin-3-*O*-glucoside. The gene was upregulated in leaves of ‘*Ca*. P. solani’-positive plants only in September (Figure 8). The *LAR* gene, involved in proanthocyanins (flavan 3-ols) synthesis, was upregulated by 3-fold in July in ‘*Ca*. P. solani’-positive plants and even more in mid-summer (16-fold), while in late summer, although overexpressed, the transcript level decreased (4-fold) (Figure 8).

#### 2.4.2. SA and Biosynthesis of Its Derivatives

With the data obtained on SA accumulation and the increase in its GA derivative, we investigated the expression of genes such as *ICS* (E.C. 5.4.4.2) and *DMR6* (E.C. 1.14.11) (Figure 9) (in addition to *PAL,* whose expression profile is presented above in Figure 7), which are responsible for SA and GA syntheses, as indicated by several authors [15,41,42,51,52].

The *ICS* gene was upregulated in ‘*Ca*. P. solani’-positive plants for the full growth season (Figure 9). The expression of *DMR6* was already high in positive plants (Figure 9) in spring and reached its highest level of overexpression in July (about 14-fold change), maintaining the upregulation in August (about 5-fold change) and September (about 2-fold change).

#### 2.4.3. Expression Patterns of SA-Dependent Defense-Related Genes

In addition, we have analyzed the expression of plant pathogenesis-related genes *PR1, PR2,* and *PR5* and their activator *NPR1* (Figure 10). The results highlighted that the *NPR1* gene was still upregulated in spring and maintained its upregulation until August. *PR1* was upregulated in grapevine leaves upon infection starting from July and reached its maximum level of overexpression in September. The *PR2* transcript showed a higher expression in positive plants in the final sampling, while *PR5* was upregulated in positive plants from August to September.

## 3. Discussion

Many studies on plant response to phytoplasma infection have analyzed the late growing season, when the disease symptoms are well evident, analyzing the metabolome and the expression of targeted genes [12,53]. Instead, in our study, we covered a complete growing season to evaluate both the early response of grapevine to ‘*Ca*. P. solani’ and the late response when disease symptoms appear.

As previously stated, typical BN symptoms on Sangiovese grapevines in central Italy are generally not evident in spring and become visible in the late summer [13]. In fact, our experiments confirmed that the leaves of ‘*Ca*. P. solani’-positive plants showed no symptoms in spring and in early summer (May and July). On the other hand, weak symptoms, such as reddish bands, appeared along the main veins in August, and gradually covered large leaf areas in September, when plants showed severe symptoms such as discoloration of veins and the reddening of laminas. A similar behavior was observed in ‘Nebbiolo’, which is less susceptible to the Flavescence dorée phytoplasma. In this variety, symptoms are evident only in mid-summer, while in ‘Barbera’, a susceptible cultivar, severe symptoms are already manifest in early summer [15]. Moreover, in ‘Manzoni Bianco’, a BN medium-susceptible cultivar, the symptoms on the leaves are delayed in late summer, with yellowing near the veins [24].

In our work, the analysis of the metabolites highlighted the presence of different phenolic compounds, regardless of health status, including SA and GA. Comparing free and conjugated forms of both phenolics, most were in the conjugated form, in accordance with previous works regarding other compatible interactions [54,55].

In our study, plant immunity seems to be exclusively related to SA and GA, because jasmonic acid, which is known to be involved in the plant response to phytoplasmas [26], was not detected through HPLC analysis in any sample (data not shown). This therefore suggests that, in the interaction between the grapevine cv. Sangiovese and ‘*Ca*. P. solani’, a key role is played by SA and its derivatives, such as GA.

‘*Ca*. P. solani’ seems to induce a high level of conjugated SA forms in infected leaves at the beginning of the growth season, which was maintained constant until the onset of severe disease symptoms, when the conjugated SA reached the maximum accumulation (Figure 4). GA in glycosylated form was constitutively present in the leaves of negative plants in all growth stages. However, in the leaves of positive plants, it gradually increased to approximatively 200 μg/g FW, which is about 13 times higher than conjugated SA when symptoms are more pronounced.

As reported in the literature, this behavior may be related to the required fine-tuning homeostasis of SA in plants. In addition to upstream regulation, an active SA level has been shown to be modulated through metabolic modifications, such as glycosylation, methylation, and hydroxylation [41]. SA hydroxylation is the major pathway for SA catabolism [31,37]. SA could therefore represent the early signaling molecule that triggers the plant response to pathogen. It was then catabolized by the hydroxylation reaction to its two principal derivatives, 2,3- DHBA and GA, which then became glycosylated (Figure 11).

We found that 2,3-DHBA was accumulated from mid-summer (Figure 5A), in line with Zhang et al. [37,40]. These authors reported that SA 3-hydroxylase, which is responsible for 2,3-DHBA biosynthesis, was more specifically expressed at the mature and senescence stage, suggesting its key role in preventing SA overaccumulation. In our experiment, 2,3-DHBA was detected in conjugated form, supporting the idea that this metabolite represents an inactive form of SA.

Most of GA (about 99%) was present in the leaves of positive plants as the conjugated form (Figure 4C). On the other hand, the maximum content of GA free-form content was detected in September, but representing less than 1% of the total (free + conjugated), and negative plants accumulated very low levels of free GA during the growth season (Figure 4A).

Although SA was also predominant in the conjugated form, in ‘*Ca*. P. solani’-positive plants, the free form was present at a higher level than in the leaves of negative plants, and it quickly increased (about threefold) to a similar level to free GA in September in positive plants (Figure 4B).

This could be explained by considering that free SA basal levels vary in different plant species (in *Arabidopsis,* the level ranges from 0.24 to 1 μg/g FW; in *Oryza sativa,* from 0.01 to 37.19 μg/g FW) [40]. Plants generally maintain SA homeostasis by fine-tuning the balance between the biosynthesis and catabolism of SA to regulate biological functions, photosynthesis, and pathogen responses [56]. Therefore, upon pathogen infection, a small amount of SA produced in planta remains in a free state in order to activate the defense response signaling. However, most SAs are subjected to biological modifications to prevent toxic accumulation as shown in *Arabidopsis*, where SA was glycosylated and then translocated to vacuole for storage [31,37]. Conjugated SA is the most common inactive form of SA, which can be transformed back into the active form when plants are attacked by pathogens [30].

Our finding agrees with the work of Zhang et al. [40], who reported that in *Arabidopsis,* free 2,3-DHBA and GA were not detected by HPLC analysis, indicating that the levels of these free acids are much lower than free SAs. In addition, according to previous findings [55,57], the conjugation of phenolics with sugar may regulate the endogenous level of free phenolics to protect plants from the toxic effects of free phenolics.

Conjugated GA accumulates in response to different types of plant pathogen interactions in much higher levels than conjugated SAs [58,59], and although the enzymes that catalyze GA glycosylation are known, the enzymes that convert SA in GA are still unknown. Some glycosyltransferases have been identified due to their function in the glycosylation of 2,3-DHBA and GA in vitro [60]. Huang et al. [41] demonstrated that the glycosylation reaction catalyzed by UGT76D1 glycosyltransferase (E.C. 2.4.1) on DHBAs plays a key role in the plant innate immune response through the modulation of SA homeostasis because UGT76D1 overexpression lines show the upregulation of the SA-responsive *PR1* and *PR2* genes. The same authors also suggested that the increase in conjugated DHBAs activates a mechanism of a positive feedback loop to induce SA biosynthesis, providing a continuous flux toward SAs and DHBAs, which occurs with an enhanced plant defense response.

SA biosynthesis requires the primary metabolite chorismate, and occurs through two major enzymatic pathways, one involving the phenylalanine ammonia lyase (PAL) pathway, and another involving a two-step process metabolized by the enzyme isochorismate synthase (ICS), which converts chorismate to isochorismate [42]. Our data showed an upregulation of the *PAL* transcript in leaves of ‘*Ca*. P. solani’-positive plants in late summer. On the other hand, *ICS* expression was higher in positive plants during all sampling stages (Figure 9), with a good correlation with SA accumulation during the growth season in the leaves of positive plants. In agreement with our results, several studies have revealed that the SA biosynthetic genes are upregulated in whole leaves of grapevines infected by ‘*Ca*. P. solani’ [23,25,26,35].

The importance of the two pathways varies in different species. In *Arabidopsis*, the ICS pathway seems to be more important for SA biosynthesis, while rice plants use the PAL pathway, and soybean activates both pathways [61]. For grapevine, we hypothesize (Figure 11) that plants react to phytoplasma infection by already activating the ICS pathway in spring, even if no symptoms are evident, preferring the ICS pathway for SA synthesis. On the other hand, the PAL upregulation registered in infected plants in late summer is probably linked to phenylpropanoid biosynthesis.

Our results on the expression profile of *DMR6* (Figure 9) showed an upregulation in infected plants already in spring, correlating with the high content of GA observed in ‘*Ca*. P. solani’-positive plants. This suggests that GA could be synthesized from SA through a reaction catalyzed by the DMR6-like protein. In accordance with our data, *DMR6* was shown to be upregulated early in BN-diseased and -recovered grapevine plants [25]. In addition, a statistical model of general plant pathology proposed by Rotter et al. [35] assumed that BN is linked to the differential expression of the *DMR6* gene in infected and uninfected plants of ‘Chardonnay’ grapevine plants. In line with Prezelj et al. [34], this suggests that *DMR6* represents a potential early marker gene in the diagnosis of BN grapevine disease. Zhang et al. [40] have proposed that DMR6, a 2-oxoglutarate-Fe (II) oxygenase, acts in *Arabidopsis* as a salicylic acid 5-hydroxylase (S5H), which converts SA in GA in vivo and in vitro. The same authors speculated that the role of DMR6 is to regulate SA homeostasis during the plant response to pathogens through a feedback mechanism induced by SA.

Although studies have highlighted its involvement in plant immunity through a direct and indirect role, the biological role of GA remains unclear. According to Belles et al. [50,58], GA represents a pathogen-inducible signal in addition to SA for the activation of plant defense, based on the accumulation of GA in conjugated form in the case of exocortis viroid (CEVd) or *Prunus necrotic ringspot virus* (PNRSV) infection in *Cucumis sativus* and *Gynura aurantica*. In addition, an increase in conjugated GA, although lower, was registered in cucumber inoculated with a low titer of *Pseudomonas syringae* pv. tomato.

Other authors [40,41,62] suggest that GA is involved in plant pathogen resistance by acting as a regulator of SA homeostasis.

To evaluate the probable role of GA in salicylic signaling function in response to phytoplasma attack, we analyzed the expression of the genes coding well-known plant pathogenesis-related proteins such as PR1, PR2, and PR5, which are considered as SA markers. We also analyzed the gene coding NPR1, a key regulator of SA-mediated signaling transduction, which is a useful molecular marker for the SAR response whose expression is salicylic acid responsive. The NPR protein could be considered a hub that controls the reprogramming of gene expression induced by SA, probably via interaction with other compounds [42,63].

Our results (Figure 9) showed that *NPR1* was upregulated in infected grapevines from early summer, before the symptoms’ appearance, thus confirming that it represents the first activator of PR protein-mediated response. NPR1 should activate PR1, which seems to be involved in the “early” response, as it was upregulated in infected plants already in July and reached maximum expression in September when symptoms were well evident. Consistently with our data, *PR1* were upregulated in ‘Chardonnay’, suggesting that in response to ‘*Ca*. P. solani infection, the SA signaling pathway is triggered [24].

According to Dermastia et al. [25], who showed that *PR2* and *PR5* genes were upregulated in the ‘*Ca*. P. solani’-infected grapevine cultivar ‘Chardonnay’, we found an upregulation of *PR2* and *PR5* in leaves of infected plants in the late summer, when symptoms appeared. Also, *PR2* and *PR5* genes are commonly considered as molecular markers for SA-dependent systemic-acquired resistance (SAR) signaling, and their expression is regulated by SA [27]. All these results suggest that ‘*Ca*. P. solani’ induces the SA-dependent response of infected Sangiovese grapevines, although we detected a higher conjugated GA level than conjugated SA in infected plants.

Therefore, in ‘Sangiovese’, as a probable regulator of the SA level, GA is somehow able to modulate the expression of *NPR1* and pathogen-related genes, *PR1*, *PR2*, *PR5*, thus confirming that signaling mediated by SA and GA is responsible for the response to BN disease (Figure 11).

We analyzed the phenylpropanoid content in both health statuses, as the flavonoid metabolism is involved in SA-related stress signaling, as reported by other authors [22]. The accumulation of flavonoids and the activation of genes involved in the flavonoid biosynthesis have also been found in multiple phytoplasma infected plants, such as paulownia, grapevine, jujube, and Mexican lime [64]. An increased flavonoid synthesis in phytoplasma-infected plants may be part of the plant’s natural defensive response against pathogen infections. HPLC ESI/MS-TOF analysis identified some phenolic acids, known as hydroxycinnamic acids (HCAs) with a significant increase in p-coumaric and ferulic acid only in symptomatic ‘*Ca*. P. solani’-positive plants. Their accumulation has been frequently observed as a result of pathogen infections [65,66]. A variation of the HCA content was also reported by Geny et al. [67] after fungal infection in grape berries. On the other hand, an accumulation of ferulic acid has been observed in *Vitis vinifera* cv. ‘Chardonnay’ BN-diseased plants in earlier phenological stages of shoot lignification [68]. The increase in p-coumaric level in September in extracts obtained from the leaves of positive plants, together with the high upregulation of the *CH4* gene, explains the role of p-coumaric acid as a precursor of many flavonoid compounds.

Our results showed higher levels of catechin and epicatechin in ‘*Ca*. P. solani’-positive plants in September. These results were complemented by transcript analysis, which underlined that in positive plants, the *LAR* transcript showed a consistent upregulation starting from July until late summer, for a longer period of time than indicated by Negro et al. [38].

Flavan-3-ols, including the monomeric catechin and polymeric proanthocyanidins (PAs), are major end products of the flavonoid biosynthetic pathway in many plant species [69]. They are involved in the protection against pathogen infection, as their biosynthesis is often induced by mechanical wounding and pathogen infections [70,71]. In poplar, their content increased in infected leaves; the transcript of *LAR* and *anthocyanidine reductase* (*ANR*) genes, involved in the last steps of biosynthesis, were upregulated upon infection [72].

An interesting result in relation to anthocyanins was that cyanidin 3-*O*-glucoside reached the highest level in ‘*Ca*. P. solani’-positive plants in the last sampling period. Because one of the typical symptoms provoked by phytoplasma is the leaf redness, the anthocyanins accumulation is likely responsible for this phenomenon [73]. In addition, we found a high level of quercetin and kaempferol in positive plants in mid and late summer, respectively. These flavonols, which are a subgroup of flavonoids, are primarily synthesized from dihydroflavonols by FLS; the trend of these compounds correlates with an increase in *FLS* expression, in infected leaves. Similarly, in grapevines infected by viruses, flavanols synthesis was enhanced by the higher expression of *FLS* than healthy ones [74]. These results confirm the assumption [13,23,74] that, upon phytoplasma infection through the activation of the NPR1 regulator, SA signaling determines a reprogramming of genes expression, which results in a modulation of phenylpropanoid biosynthetic pathway genes, which are generally upregulated in infected plants that show symptoms.

Our data suggest that SA represents the early signaling molecule that triggers several pathways involved in the ‘Sangiovese’ physiological response to BN, culminating with disease development. This mechanism could provide Sangiovese with a partial resistance to BN disease, although it is unable to completely combat it.

We proposed a hypothetical working model for the signal mechanisms mediated by SA in the interaction of Sangiovese grapevine–phytoplasma during the growth season (Figure 11). When plants are challenged by phytoplasma, this triggers plant hormone regulation, which significantly increases SA biosynthesis. SA immediately activates a signaling pathway in the host, mainly consisting of the reprogramming of different physiological mechanisms, through the induction of some phenylpropanoids biosynthetic and SA-responsive genes, such as *NPR1*, *PR1*, *PR2,* and *PR5*.

When SA is accumulated at a low basal level (May), a gene coding for SA hydroxylation, such as *DMR6*, was induced in infected leaves to prevent SA accumulation, thus increasing the GA content. Most of the GA produced was quickly glycosylated, and the continuous accumulation of glycosylated GA probably triggers an unknown positive feedback mechanism to synthetize SA, in line with the model proposed by Huang et al. [41]. This mechanism ensures a flux of SAs and DHBAs to maintain the activation of the plant response to phytoplasma, and at the same time, limiting SA toxicity.

In support of this hypothesis, a constant upregulation of *ICS* was detected during the growth season; moreover, although we did not analyze a grapevine glycosyltransferase, SA, GA, and 2,3-DHBA were found in the conjugate form throughout the experimental period, confirming the important role of glycosylation.

## 4. Materials and Methods

### 4.1. Plant Samples and Phytoplasma Detection

Field surveys were conducted in a cv. Sangiovese vineyard (*Vitis vinifera*, L. ‘Sangiovese’ I-SS F9 A5 48) located in Greve in Chianti (Tuscany, central Italy) where ‘*Ca*. P. solani’-positive and -negative plants were detected through multi-year monitoring. Leaf sampling was carried out at different periods according to symptom appearance from late spring to late summer (May, July, August, and September), following the gradual appearance of symptoms in positive plants. Sampling was always performed on the same plants all the time, collecting 10–15 leaves from five ‘*Ca*. P. solani’-positive plants and five ‘*Ca*. P. solani’-negative plants. In all sampling periods, the severity of symptoms was classified according to a grapevine reddening symptomatic scale from 0 to 3, as reported by Pierro et al. (2018): (i) symptom severity class 0 = plants with no symptoms, (ii) symptom severity class 1 = one shoot with mild leaf symptoms, (iii) symptom severity class 2 = two to three shoots with leaf symptoms, and (iv) symptom severity class 3 = more than three shoots with symptoms of reddening leaf and berry shrivel.

The collected leaves were stored at −20 °C until DNA extraction for phytoplasma detection or were lyophilized (Christ alpha 2-4 LSC plus, Osterode am Harz, Germany) for biochemical analysis. The DNA was extracted following the procedure described in Nicolì et al. [75]. Specific detection of ‘*Ca*. P. solani’ was carried out by amplification of 16S ribosomal DNA using a TaqMan assay following reaction conditions as described in Angelini et al. [76]. A threshold cycle of <37 was associated with the presence of ‘*Ca*. P. solani’.

Both ‘*Ca*. P. solani’-positive and ‘*Ca*. P. solani’-negative plants were tested for some of the most common viruses of *Vitis* spp. (European Commission directive 2005/43/EC). Diagnostic tests (real-time PCR) were carried out for *grapevine fanleaf virus*, *arabis mosaic virus*, *grapevine leafroll*-*associated virus 1*, *grapevine leafroll-associated virus 3*, and *Grapevine fleck virus* [77,78,79]. Both healthy and infected samples were collected from plants that had negative results in all diagnostic tests. In addition, the protection of ‘*Ca*. P. solani’-positive and ‘*Ca*. P. solani’-negative plants was carried out according to common practices in the area, and sampled plants showed no symptoms related to *Uncinula necator* (Schw.) Burr., *Plasmopara viticola* (Berk. et Curt.) Berl. et de Toni, *Botrytis cinerea* Pers, and *Guignardia bidwellii* (Ellis) Viala and Ravaz.

### 4.2. Extraction of Phenylpropanoids

Samples (about 0.5 g of tissue) were ground in a pre-cooled mortar to a fine powder using liquid nitrogen and then homogenized in 1.5 mL water/methanol 40/60 *v*/*v*. The extracts were centrifuged for 15 min at 10,000× *g* to remove cellular debris. The supernatant for each sample was divided into two equal aliquots and vacuum-dried at 40 °C. An aliquot was resuspended in 900 μL of 50 M^−3^ sodium acetate buffer (pH 4.5) and 100 μL of water to analyze free SA and GA together with phenolic compounds. Another aliquot was resuspended in 900 μL of 50 M^−3^ sodium acetate (pH 4.5) and 100 μL of water containing 10 U of almond β-glucosidase (3.2.1.21) (14.3 U mg^−1^, Sigma Aldrich, Milano, Italy) to remove sugars bound to phenolic compounds. The enzymatic reaction was incubated overnight at 37 °C and then stopped by adding 75 µL of water/perchloric acid 30/70 *v*/*v* to the incubation mixture and stored at 4 °C for 1 h. After centrifugation at 14,000× *g* for 15 min to remove polymers, the supernatants were extracted with 2.5 mL of cyclopentane/ethyl acetate 1/1 *v*/*v*. The organic upper phase was collected and dried at 40 °C. The residue was resuspended in 200 μL of methanol and filtered through 0.22 μm prior to HPLC analysis.

### 4.3. HPLC ESI/MS-TOF Analysis

The phenolic characterization and quantification of leaf extracts were performed using an Agilent 1200 High Performance Liquid Chromatography (HPLC) System (Agilent Technologies, Palo Alto, CA, USA) equipped with a standard autosampler and an Agilent Zorbax Extend-C18 analytical column (5 × 2.1 cm, 1.8 μm), as reported by Vergine et al. [80]. The HPLC system was coupled to an Agilent diode-array detector (wavelength 280 nm) and an Agilent 6320 TOF mass spectrometer equipped with a dual ESI interface (Agilent Technologies) operating in negative ion mode. Detection was carried out within a mass range of 50–1700 *m*/*z*. Accurate mass measurements of each peak from the total ion chromatograms (TICs) were obtained by an ISO Pump (Agilent G1310B) using a dual nebulizer ESI source that introduces a low flow (20 μL min^−1^) of a calibration solution containing the internal reference masses at *m*/*z* 112.9856, 301.9981, 601.9790, and 1033.9881, in negative ion mode. The anthocyanins were identified with the same method, but with positive ionization (detection wavelength 280 and 520 nm), using the internal reference masses at *m*/*z* 121.050873, 149.02332, 322.048121, and 922.009798, as reported by Aprile et al. [81].

The compounds were quantified using calibration curves of authentic standards (salicylic acid, gentisic acid, 2,3-DHBA, catechin, epicatechin, quercetin, kaempferol 3-*O*-glucoside, resveratrol glucoside, cyanidin 3-glucoside).

### 4.4. RNA Extraction and qRT-PCR

Leaf tissues sampled from healthy and infected plants at four growth stages were frozen in liquid nitrogen, and total RNA was isolated from 0.1 g of samples using TRIZOL (Invitrogen, Carlsbad, CA, USA). cDNA synthesis was carried out using TaqMan^®^ Reverse Transcription Reagents (Applied Biosystems, Foster City, CA, USA) according to the manufacturer’s standard protocol. Amplification reactions were performed using the Applied Biosystems^®^ QuantStudio^®^ 3 Real-Time PCR System. Each reaction consisted of 2 ng of cDNA, 12.5 μL of Power SYBR Green RT-PCR Master mix (Applied Biosystems), 5.0 M-6 forward and reverse primers, and ultrapure DNase/RNase-free water (Carlo Erba Reagents S.r.l.) in a total volume of 25 μL. The cycling conditions were as follows: 2 min at 50 °C and 10 min at 95 °C, followed by 45 cycles of 95 °C for 15 s, and 60 °C for 1 min. Melting curve analysis was performed after PCR to evaluate the presence of non-specific PCR products and primer dimers.

The primers (Appendix A) were designed with Primer Express Software 3.0 on the mRNA sequences obtained from the literature [13,23,34,74].

Quantitative real-time PCR was used for rapid and reliable quantification of mRNA transcription. However, selecting an appropriate reference gene is crucial for an exact comparison of mRNA transcription in different samples. Of the various genes reported in the literature, we used *COX* (*cytochrome c oxidase*, E.C. 1.9.3.1) as a reference gene, as reported by Bertazzon et al. [82].

For the relative quantification of gene expression, we calculated the fold changes (FCs) using the following formula:F= 2^(−∆∆CT)^
where:∆∆CT = [(CT _target gene_) − (CT _reference gene_)]_positive sample_ − [(CT _target gene_) − (CT _reference gene_)] _healthy sample_

The CT data are expressed as the average of 12 samples.

For each individual sample, four technical replicates were analyzed.

With the cut-off value of a 2-fold change in gene expression, we considered the upregulation in positive plants compared to negative plants.

### 4.5. Statistical Analysis

All data were reported as the mean ± SD in triplicate for each analyzed sample (n = 5, ‘*Ca*. P. solani’-positive and -negative plants, respectively). The statistical analysis was performed using multiple *t*-tests (FDR = 5%) to highlight the differences between ‘*Ca*. P. solani’-positive and -negative leaves for each physiological parameter analyzed. Statistical analyses were performed using GraphPad v. 6.01. A one-way ANOVA test was applied to expression gene data.

## 5. Conclusions

Our results confirm the previous evidence that phytoplasma interact with the SA pathway [36]. In addition, we identified a high level of conjugate gentisic acid in grapevine infected by *Candidatus* Phytoplasma solani.

The altered physiological conditions may also be a consequence of the phytoplasma effects on development and stress signaling pathways and of the interactions between them.

The biology of phytoplasmas and the actual defense mechanisms of plants are still unknown because the pathogens need plants and insects for survival in nature, which means that an “in vitro” cultivation is very difficult. However, the metabolomic and transcriptomic data that we obtained confirm that innate immunity, phytohormone signaling, and many phenylpropanoid compounds, which constitute a complex defense network in plants, are involved in the response of grapevine-to-phytoplasma infection.

Although SA is essential in the grapevine–phytoplasma interaction, it is not the exclusive signal, with GA appearing to play a role in enabling or modulating the grapevine response to phytoplasma infection. GA did not interfere with the biological effects of SA; however, it represents a component of the grapevine SA-dependent response, probably for the role to fine-tune the SA level.

Our results suggest a new point of view concerning the physiological mechanisms underlying phytoplasma–grapevine interactions aimed at improving disease control strategies.

## Figures and Tables

**Figure 1 plants-12-02695-f001:**
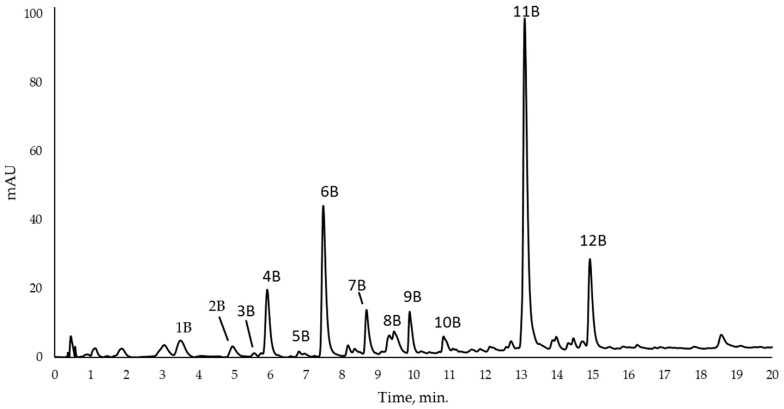
Typical chromatogram recorded at 280 nm of *Vitis vinifera* leaf sample extract after digestion with β-glucosidase for the identification of phenylpropanoid compounds. The identification of single peak is reported in Table 2.

**Figure 2 plants-12-02695-f002:**
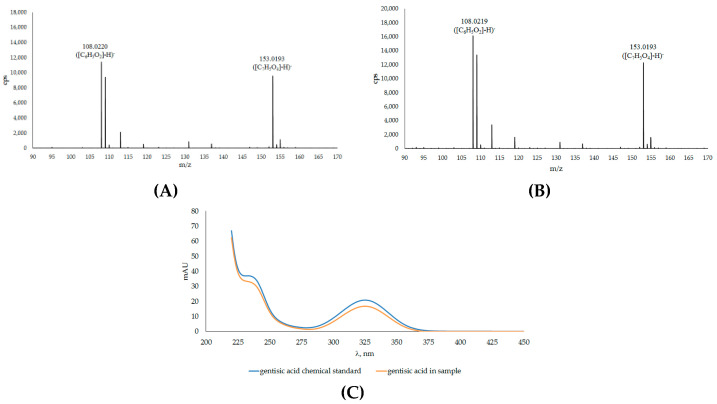
Mass spectra of 2,5 dihydroxybenzoic acid (gentisic acid, GA) found in the sample (**A**) and of the chemical standard (**B**) and their UV/Vis absorption spectrum of the peak 1B shown in Figure 1 after sample digestion with β-glucosidase (**C**).

**Figure 3 plants-12-02695-f003:**
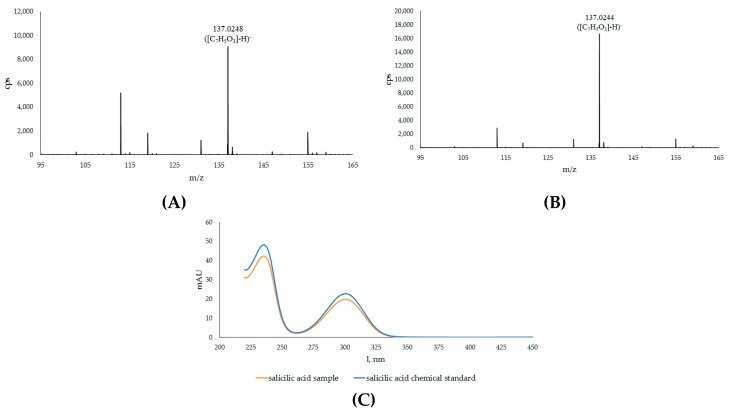
Mass spectra of the salicylic acid found in the sample (**A**) and of the chemical standard (**B**) and their UV/Vis absorption spectrum of the peak 8B shown in Figure 1 after sample digestion with β-glucosidase (**C**).

**Figure 4 plants-12-02695-f004:**
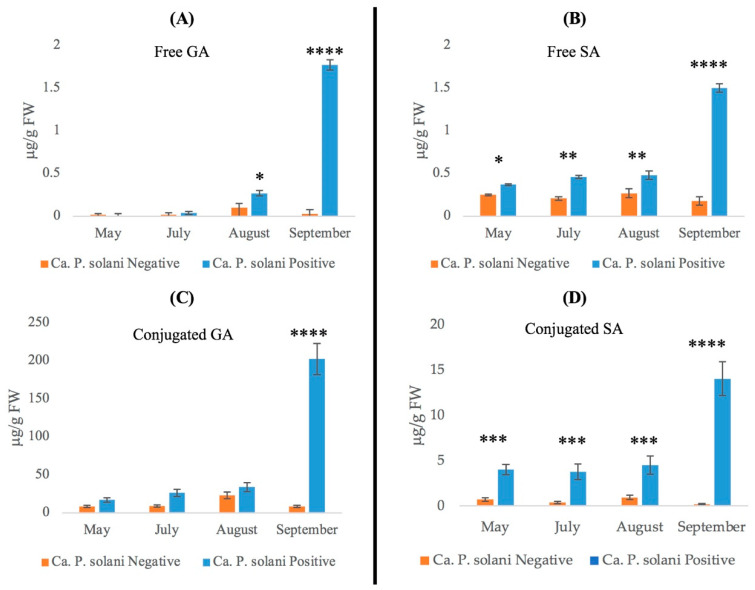
Quantification of gentisic acid (GA) and salicylic acid (SA) obtained from grapevine leaves collected from ‘*Ca*. P. solani’-negative or -positive plants at different growth stages: (**A**) free GA, (**C**) conjugated gentisic acid, (**B**) free SA, (**D**) conjugated salicylic acid. The statistical analysis between Ca. P. solani-negative and Ca. P. solani-positive leaves was carried out using a multiple *t*-test (FDR = 5%) and significant differences are marked by asterisks: * *p* < 0.05, ** *p* < 0.01, *** *p* < 0.001, **** *p* < 0.0001. Values are reported as means and standard deviation of five harvested samples (*n* = 5 ‘*Ca*. P. solani’-negative and ‘*Ca*. P. solani’-positive plants, respectively), each measured in three technical replicates.

**Figure 5 plants-12-02695-f005:**
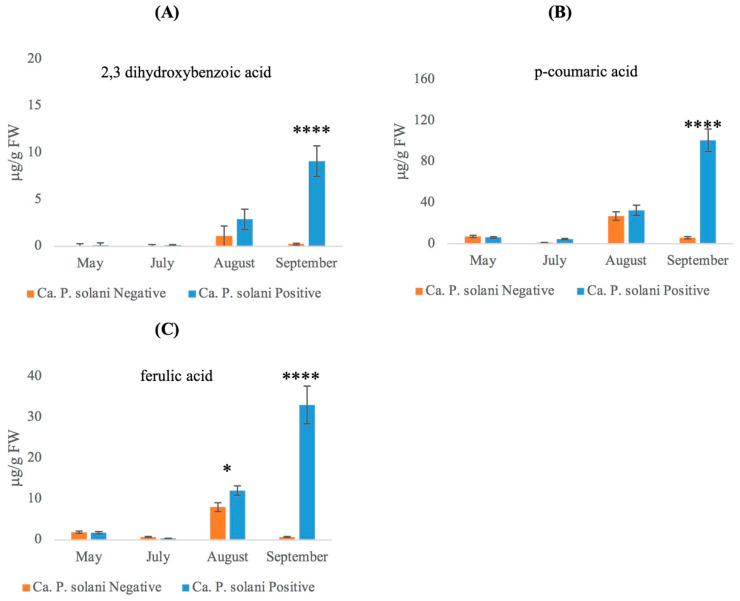
Accumulation of phenolic compounds (**A**) 2,3 dihydroxybenzoic acid; (**B**) p-coumaric acid; (**C**) ferulic acid after digestion with β-glycosidase of samples obtained from grapevine leaves collected from ‘*Ca*. P. solani’-negative or -positive plants at different growth stages. The statistical analysis between leaves collected from ‘*Ca*. P. solani’-positive and -negative plants carried out using a multiple *t*-test (FDR = 5%) and significant differences are marked by asterisks: * *p* < 0.05, **** *p* < 0.0001. Values are reported as means and standard deviation of five harvested samples (*n* = 5 ‘*Ca*. P. solani’-negative and ‘*Ca*. P. solani’-positive plants, respectively), each measured in three technical replicates.

**Figure 6 plants-12-02695-f006:**
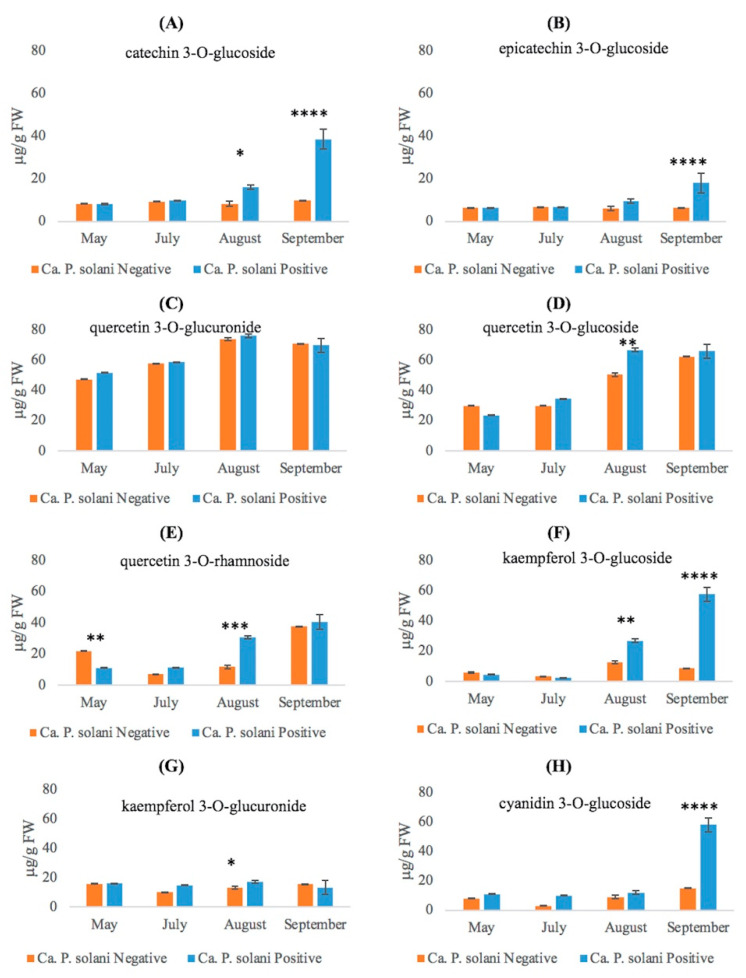
Quantitative determination of flavonoid compounds (**A**) catechin 3-*O*-glucoside; (**B**) epicatechin 3-*O*-glucoside; (**C**) quercetin 3-*O*-glucoronide; (**D**) quercetin 3-*O*-glucoside; (**E**) quercetin 3-*O*-rhamnoside; (**F**) kaempferol 3-*O*-glucoside; (**G**) kaempferol 3-*O*-glucuronide; and (**H**) cyanidin 3-*O*-glucoside in leaves collected from ‘*Ca*. P. solani’-positive and -negative plants at different growth stages. The statistical analysis between Ca. P. solani-negative and Ca. P. solani-positive leaves was carried out using a multiple *t*-test (FDR = 5%) and significant differences are marked by asterisks: * *p* < 0.05, ** *p* < 0.01, *** *p* < 0.001, **** *p* < 0.0001. Values are reported as means and standard deviation of five harvested samples (*n* = 5 ‘*Ca*. P. solani’-negative and ‘*Ca*. P. solani’-positive plants, respectively), each measured in three technical replicates.

**Figure 7 plants-12-02695-f007:**
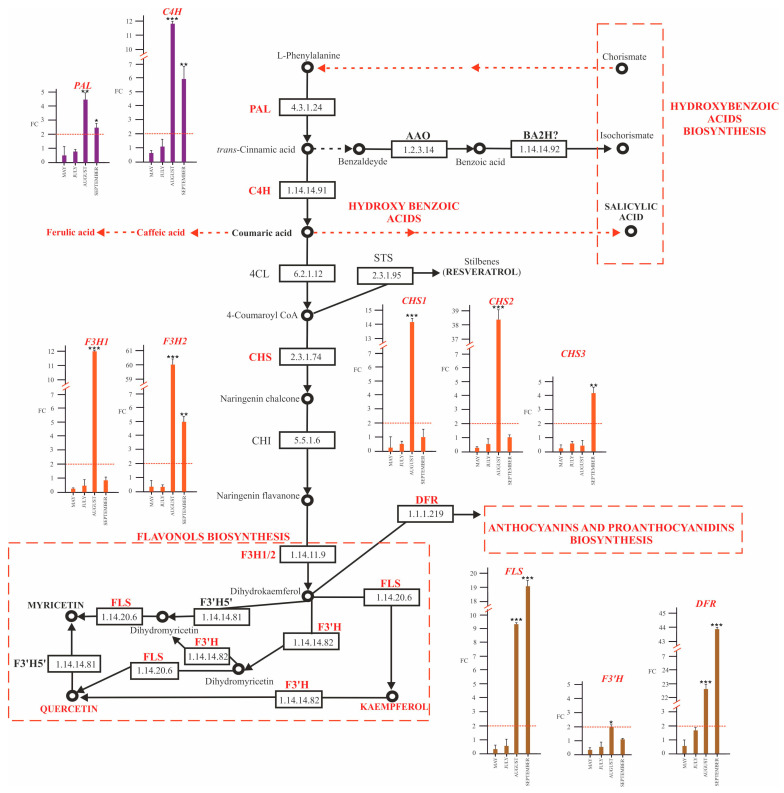
General phenylpropanoid biosynthetic pathway and expression analysis by qRT-PCR of biosynthetic genes in leaves collected from ‘*Ca*. P. solani’-positive plants, expressed as fold change (FC) relative to ‘*Ca*. P. solani’-negative plants. Samples were collected in May, July, August, and September. The reported compounds detected by HPLC ESI/MS-TOF and key biosynthetic genes and relative enzymes are in red. PAL, phenylalanine ammonia-lyase; C4H cinnamate 4-hydroxylase; CHS, chalcone synthase; F3H, flavanone-3-hydroxylase; F3′H, flavonoid-3′-hydroxylase; FLS, flavonol synthase; DFR, dihydroflavonol reductase. Red lines highlight the 2-fold change to graphically identify genes upregulated in positive plants compared to negative ones. ANOVA results were reported based on their statistical significance. * *p* < 0.05, ** *p* < 0.01, *** *p* < 0.001.

**Figure 8 plants-12-02695-f008:**
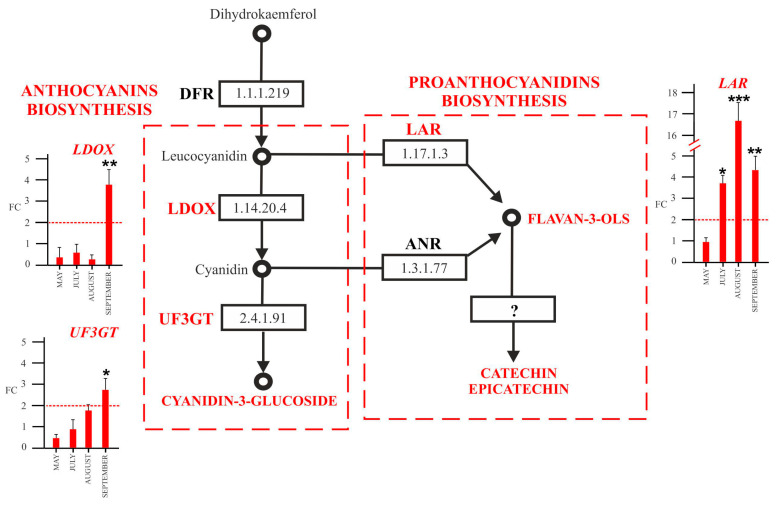
Key steps of anthocyanin and proanthocyanidin biosynthetic pathways and expression analysis by qRT-PCR of genes involved in their biosynthesis in leaves collected from ‘*Ca*. P. solani’-positive plants, expressed as fold change (FC) relative to ‘*Ca*. P. solani’-negative plants. Samples were collected in May, July, August, and September. The reported compounds detected by HPLC ESI/MS-TOF and the key biosynthetic genes and relative enzymes are in red. LDOX, leucoanthocyanidin dioxygenase; UF3GT, UDP-glucose flavonoid 3-*O*-glucosyltransferase; LAR, leucoanthocyanidin reductase. Red lines highlight the 2-fold change to graphically identify genes upregulated in positive plants compared to negative ones. ANOVA results were reported based on their statistical significance. * *p* < 0.05, ** *p* < 0.01, *** *p* < 0.001.

**Figure 9 plants-12-02695-f009:**
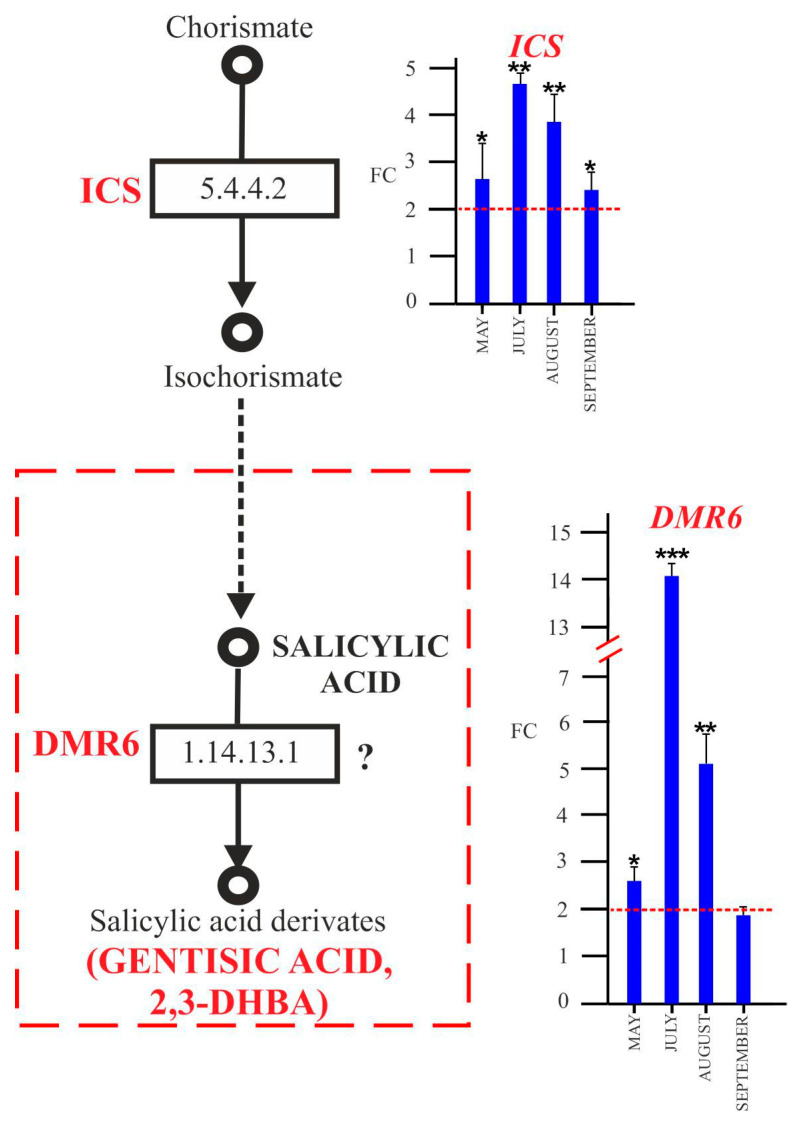
Key steps of hydroxybenzoic acid biosynthetic pathway and expression analysis by qRT-PCR of genes involved in their biosynthesis in leaves collected from ‘*Ca*. P. solani’-positive plants, expressed as fold change (FC) relative to ‘*Ca*. P. solani’-negative plants. Samples were collected in May, July, August, and September. The reported compounds detected by HPLC ESI/MS-TOF and the key genes and relative biosynthetic enzymes are in red. *ICS,* isochorismate synthase; *DMR6*, downy mildew-resistant 6. Red lines highlight the 2-fold change to graphically identify genes upregulated in positive plants compared to negative ones. ANOVA results were reported based on their statistical significance. * *p* < 0.05, ** *p* < 0.01, *** *p* < 0.001.

**Figure 10 plants-12-02695-f010:**
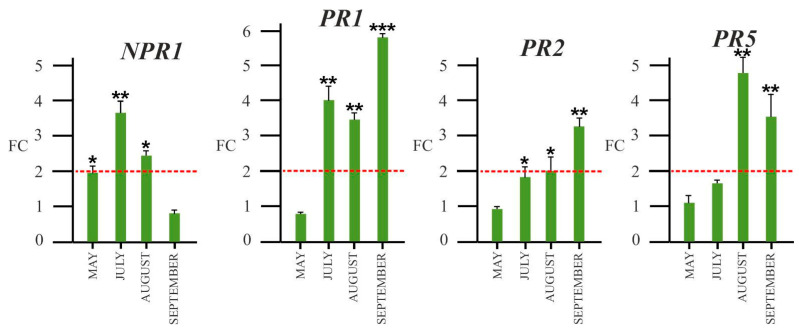
Expression analysis by qRT-PCR of genes coding NPR1 (non-expressor of pathogenesis-related genes 1) and pathogen-related proteins (PRs) in leaves collected from ‘*Ca*. P. solani’-positive plants, expressed as fold change (FC) relative to ‘*Ca*. P. solani’-negative grapevine leaves sampled, respectively, in May, July, August, and September during a growth season. *NPR1:* Non-expressor of pathogenesis-related protein1, *PR1, PR2, PR3:* pathogenesis-related protein 1,2,5. Red lines highlight the 2-fold change to graphically identify genes upregulated in positive plants compared to negative. ANOVA results were reported based on their statistical significance. * *p* < 0.05, ** *p* < 0.01, *** *p* < 0.001.

**Figure 11 plants-12-02695-f011:**
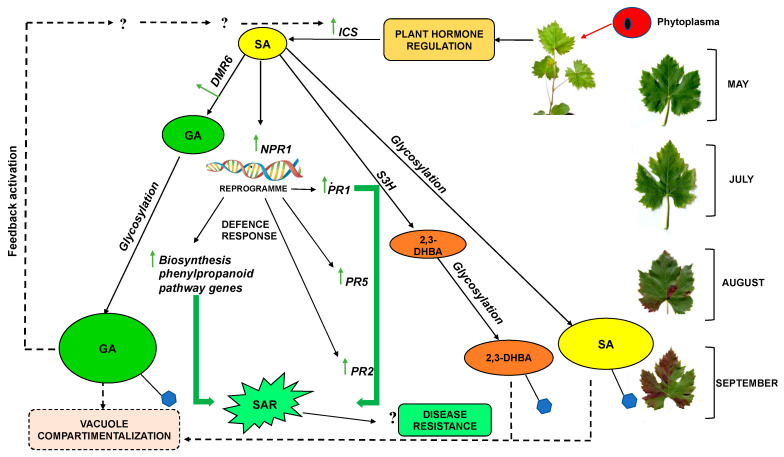
Proposed scheme for the activation of SA signaling pathways and the regulation of SA accumulation during the infection of Sangiovese grapevine by ‘*Ca*. P. solani’. SA, free salicylic acid; GA, free gentisic acid; 2,3-DHBA, free 2,3 dihydroxybenzoic acid; S3H, salicylic acid 3- hydroxylase; *DMR6*, *downy mildew resistant 6* gene; *ICS*, isochorismate synthase gene; *NPR1, non-expressor of pathogenesis-related protein1*, *PR1, PR2*, *PR5, pathogenesis-related (1, 2, 5)* genes. Sugar molecule is represented in blue.

**Table 2 plants-12-02695-t002:** List of compounds identified from leaves collected from *Vitis vinifera* cv. Sangiovese ‘*Ca*. P. solani’-positive and -negative plants after digestion with β-glucosidase. All compounds were verified by a comparison with an authentic chemical standard.

No.	Compound	RT ^a^ (min)	*m/z* exp. ^b^	*m/z* calc. ^c^	(M-H)^−^	Error (ppm)
**1B**	2,5 Dihydroxybenzoic acid (Gentisic acid)	3.711	153.0194	153.0193	C_7_H_5_O_4_	−0.65
**2B**	2,3 Dihydroxybenzoic acid	5.102	153.0196	153.0193	C_7_H_5_O_4_	−1.55
**3B**	Catechin	5.881	289.0728	289.0718	C_15_H_13_O_6_	−3.45
**4B**	Methyl benzoate	5.997	135.0468	135.0452	C_8_H_7_O_2_	−11.84
**5B**	Epicatechin	6.874	289.0723	289.0718	C_15_H_13_O_6_	−1.72
**6B**	p-Coumaric acid	7.785	163.0410	163.0401	C_9_H_7_O_3_	−5.52
**7B**	Ferulic acid	8.845	193.0514	193.0506	C_10_H_9_O_4_	−4.14
**8B**	*o*-hydroxybenzoic acid (Salicylic acid)	9.756	137.0259	137.0255	C_7_H_5_O_3_	−2.91
**9B**	Quercetin 3-*O*-glucuronide	9.952	477.0653	477.0675	C_21_H_17_O_13_	4.61
**10B**	Myricetin	10.869	317.0316	317.0303	C_15_H_9_O_8_	−4.10
**11B**	Quercetin	13.231	301.0355	301.0354	C_15_H_9_O_7_	−0.33
**12B**	Kaempferol	15.023	285.0395	285.0404	C_15_H_9_O_6_	3.15

^a^ Retention time (min), ^b^ *m*/*z* experimental, ^c^ *m*/*z* calculated from software Agilent Mass Hunter 7.0.

## Data Availability

Not applicable. All results are included in the article.

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
