# Peer review of "Signaling Cross-Talk between Salicylic and Gentisic Acid in the ‘*Candidatus* Phytoplasma Solani’ Interaction with Sangiovese Vines"

_plants, 2023, doi:10.3390/plants12142695_

Round 1

Reviewer 1 Report

Nutricati et al. investigated the molecular basis of “Bois noir” disease in grapevine by using a metabolite analysis that was further supported by expression analysis of selected genes known to be involved in plant immunity and secondary compound biosynthesis.

The results reported are interesting and support the roles of salicylic acid and gentisic acid in grapevine immune responses.

I would just recommend the authors to present statistics of the mRNA quantification results shown in Fig. 6-10, i.e. as presented for metabolite quantification figures.

Clarity of some parts of the abstract and main text needs to be improved, e.g. "Metabolomic data corroborated by gene expression analysis indicated that phenylpropanoid biosynthetic and salicylic acid responsive." is unclear.

Author Response

  • In the abstract we complete and clarify the sentence “Metabolomic data corroborated……”
  • We added statistical data, about the gene expression, in the figures 7, 8, 9, 10. We added the reference to statistical analysis in the captures of the figures 7, 8, 9, 10.

Reviewer 2 Report

Signaling cross-talk between salicylic and gentisic acid in the ‘Candidatus Phytoplasma solani’ interaction with Sangiovese vines 

Eliana Nutricati, Mariarosaria De Pascali, Carmine Negro, Piero Attilio Bianco, Fabio Quaglino, Alessandro Passera, Roberto Pierro, Carmine Marcone, Alessandra Panattoni, Erika Sabella, Luigi De Bellis, Andrea Luvisi

The authors of this manuscript studied the signaling cross-talk between SA and GA when plants are infected with phytoplasma during the season of growth. Metabolomic and qPCR analysis allows them to propose a model of how the plant responds to infection.

The article is well-written and the research is well-designed.

 Recommendations for Authors (will be shown to authors)

The following questions do not substitute for specific comments made for authors. Please give further details in the comments for authors box below.

Yes

Can be improved

Must be improved

Not applicable

Does the introduction provide sufficient background and include all relevant references?

Is the research design appropriate?

Are the methods adequately described?

Are the results clearly presented?

Are the conclusions supported by the results?

* English language and style

 Extensive editing of English language and style required

 Moderate English changes required

 English language and style are fine/minor spell check required

 I don't feel qualified to judge about the English language and style

Recommendations for Editors (will not be shown to authors)

If you answered yes to any of the following questions, please give details in the comments for editors box below.

Yes

No

Do you have any potential conflict of interest with regards to this paper?

Did you detect plagiarism?

Did you detect inappropriate self-citations by authors?

Do you have any other ethical concerns about this study?

Ratings

High

Average

Low

No Answer

* Originality / Novelty

* Significance of Content

* Quality of Presentation

* Scientific Soundness

* Interest to the readers

* Overall Merit

Comments

* Comments and Suggestions for Authors
(will be shown to authors)

Please see the annex.

Word/PDF/Zip/Rar/7z

Comments for Editors
(will not be shown to authors)

Overall Recommendation

* Overall Recommendation

 Accept in present form

 Accept after minor revision (corrections to minor methodological errors and text editing)

 Reconsider after major revision (control missing in some experiments)

 Reject (article has serious flaws, additional experiments needed, research not conducted correctly)

Comments for authors can be made available to other reviewers. Your identity will remain confidential.

Submission Date

Some specific comments are listed below, and the line number in which they are found.

Plant – phytoplasma, defense –related, must be Plant-phytoplasma, defense-related

This reviewer suggests using “plant growth regulator” instead of “plant hormones” SA, JA, and ET are not hormones.

Kaempferol 3-O glucuronide, yanidin 3-O-glucoside. The –O- must be writte in italic.

The language of the manuscript is good. This reviewer recommends small changes in the redaction of the paper as those illustrated in the abstract.

Abstract: Abstract: “Bois noir” disease associated with ‘Candidatus hytoplasma solani’ seriously ompromises the production and survival of grapevines (Vitis vinifera L.) in Europe. Understanding the plant response to phytoplasmas should help to improve disease control strategies. Using a combined metabolomic and transcriptomic analysis, this work, therefore, investigated the phytoplasma–grapevine interaction in red cultivar Sangiovese in the vineyard over four seasonal growth stages (from late spring to late summer), comparing leaves from healthy and infected grapevines (symptomatic and symptomless). We found an accumulation of both conjugate and free salicylic acid (SA) in the leaves of ‘Ca. P. solani’-positive plants from early stages of infection, when plants are still asymptomatic. For the first time, a A strong accumulation of gentisic acid (GA) associated with symptoms progression was found for the first time. A detailed analysis of phenylpropanoids revealed a significant accumulation of hydroxycinnamic acids, flavonols, flavan 3-ols, and anthocyanin cyanidin 3-O-glucoside, which are extensively studied due to their involvement in the plant response to various pathogens. Metabolomic data corroborated by gene expression analysis indicated that phenylpropanoid biosynthetic and salicylic acid- responsive.

Minor editing of English language

Author Response

  • In the abstract we have improved the language.
  • We change the “O” in italic in the name of compound Kaempferol- O- glucuronide……
  • We eliminate the space between the words “plant – phytoplasma and defense – related”
  • We change “plant hormones” with “plant growth regulators”.